# Developing Open Source Educational Resources
# for Machine Learning and Data Science

## Abstract

Education should not be a privilege but a common good. It should be openly accessible to everyone, with as few barriers as possible; even more so for key technologies such as Machine Learning (ML) and Data Science (DS). Open Educational Resources (OER) are a crucial factor for greater educational equity. In this paper, we describe the specific requirements for OER in ML and DS and argue that it is especially important for these fields to make source files publicly available, leading to Open Source Educational Resources (OSER). We present our view on the collaborative development of OSER, the challenges this poses, and first steps towards their solutions. We outline how OSER can be used for blended learning scenarios and share our experiences in university education. Finally, we discuss additional challenges such as credit assignment or granting certificates.

## 1. Introduction

Education is of paramount importance to overcome social inequalities. Globalization and broad access to the internet provide a major opportunity for allowing many more people from all over the world to access high quality educational resources. We endorse the vision of the Open Education Global initiative (OEG, a) and believe that teaching materials developed with the support of public financial resources should be openly accessible for the general public. The initiative aims at providing Open Educational Resources (OER) (OEG, b) which come with the '5R' permissions to retain, reuse, revise, remix, and redistribute the material. The UNESCO strongly promotes the concept of OER (UNESCO), has held two world congresses on OER, in 2012 and 2017, and finally adopted a recommendation on OER (UNESCO, 2019). This recommendation is claimed to be

[1]Anonymous Institution, Anonymous City, Anonymous Region, Anonymous Country. Correspondence to: Anonymous Author <anon.email@domain.com>.

Preliminary work. Under review by the Teaching Machine Learning Workshop at ECML 2021. Do not distribute.

'the only existing international standard-setting instrument on OER' (UNESCO).

A variety of Massive Open Online Courses (MOOCs) have been created in the fields of Machine Learning (ML) and Data Science (DS). These MOOCs are mostly offered by commercial platforms (e.g., Ng, 2021; Boitsev et al., 2021; Malone & Thrun, 2021; Google, 2021; Sulmont et al., 2021; Eremenko & de Ponteves, 2021) but also by university-owned platforms (e.g., MIT, 2021).

Although the material itself is often freely accessible, access to the sources that are needed to reproduce, modify and reuse the material is usually not provided. Only a small fraction of courses in ML and DS actually share all their sources, such as slides sources in .pptx or LaTeX and source codes for plots, examples, and exercises. We call those 'Open Source Educational Resources' (OSER) to underline this feature; positive examples include Montani (2021); Çetinkaya-Rundel (2021a;b); Vanschoren (2021).

Direct benefits of OSER from the perspective of lecturers include: 1. The material will often be of higher quality if additional experts are able to contribute and improve the material. 2. It is more efficient to develop a new course since material can be adapted and re-used from previously created courses legally. 3. Starting from an established OSER course, lecturers can focus on developing additional chapters and tailoring existing material to their audience.

We believe there will never be the one and only course on a certain topic – in fact, diversity in how topics are taught is important. Reasons include different constraints on the volume of a course defined by an institution's curriculum, different backgrounds (in the context of ML and DS courses, e.g., statistics, mathematics, computer science), different types of institutions (e.g. university vs. continuing education, undergraduate vs. (post)graduate), different substantive focus, and sometimes simply different styles. On the other hand, it seems natural that teachers for many subjects should be able to find networks of peers among which a considerable amount of content can be similar and we advocate that it is only sensible and efficient to share, reuse and collaboratively improve teaching resources in such cases.

But if such a network of peers or shared interest in a certain topic is established, usage of the material is often more complicated and less straight-forward, as adaptions and modifi-

cations are often still necessary to accommodate the different contexts and constraints of each institution the teachers work at. The easier and more natural such (reasonable and common) modifications are possible, the more likely it is that like-minded developers and teachers can agree to form a networked team. In our view, there are six different use cases (UC) in this context that can be classified as *usage* of the material and *contribution* to the material. Usage consists of (UC1) usage of material without any modifications for teaching, (UC2) usage of a subset of the material for a smaller course, (UC3) development of a somewhat different course where the existing material is used as a starting point. Contribution consists of (UC4) correcting errors, (UC5) improving the existing material, and (UC6) adding new material which leads to a larger OSER collection.

In our experience, sharing teaching material is much more complex than just publishing lecture slides on the web. In this paper, we describe which core principles of developing OSER in general and, specifically, for teaching ML and DS should be considered, share our experiences of applying them and point towards open questions and challenges.

## 2. OSER and Machine Learning

In this article, we discuss OSER from the perspective of teaching ML and DS. Standard material in ML and DS naturally includes theoretical components introducing mathematical concepts, methods and algorithms, typically presented via lecture slides and accompanying videos, as well as practical components such as code demos, which are important to allow for hands-on experience. In contrast to many other disciplines: 1. ML's strong foundation in statistics and algorithms allows to define and illustrate many concepts via (pseudo-)code; 2. Large, open data repositories (such as OpenML (Vanschoren et al., 2013) or ImageNet (Deng et al., 2009)) allow students to obtain hands-on experience on many different applications. 3. Many state-of-the-art ML and DS packages (such as scikit-learn (Pedregosa et al., 2011), mlr3 (Lang et al., 2019), caret (Kuhn et al., 2008), tensorflow (Abadi et al., 2015) or pytorch (Paszke et al., 2019)) are open-source and freely available so that students can directly learn to use libraries and frameworks that are also relevant in their future jobs; 4. Many concepts, algorithms, data characteristics and empirical results can be nicely visualized, and this often happens through short coded examples using the mentioned ML toolkits and open data repositories. 5. Gamification via competitions (Kapp, 2012) is possible since running experiments is (comparably) cheap, datasets are available and existing platforms, such as Kaggle InClass, provide the necessary infrastructure and have shown improved learning outcomes (Polak & Cook, 2021).

The following recommendations should always be seen in view of the points mentioned above. They also demonstrate that our community is closely connected to the open source spirit and transferring concepts related to open source to teaching in ML should feel even more natural than in other sciences. Furthermore, as students are (or better: should be) used to working with practical ML notebooks on open data sets, each source example used in a lecture chapter (to generate a plot, animation or toy result) provides a potential starting point for further student exploration.

## 3. Developing OSER – the Core Principles

We argue that developing OSER has several benefits for students as well as for lecturers and that a lot can be gained from transferring concepts and workflows from software engineering in general and open source software development specifically to the development of OSER, e.g., collaborative work in decentralized teams, modularization, issue tracking, change tracking, and working in properly scheduled release cycles. In the following we list major core principles which in our opinion provide the basis for successful development of open source resources, including brief hints regarding useful technical tools and workflows (many, again, inspired from open source software engineering) and briefly discuss connected challenges.

**Develop course material collaboratively with others.** When several experts from a specific field come together to develop a course, there is a realistic chance that the material will be of higher quality, more balanced, and up-to-date. Furthermore, the total workload for each member of the collaboration is smaller compared with creating courses individually. However, developing a course together comes with the necessity of more communication between the members of the group, e.g., to ensure a consistent storyline and a set of common prerequisites, teaching goals, and mathematical notation. In order to reduce associated costs, an efficient communication structure and the right toolkits are vital, e.g., Git for version control and Mattermost as a communication platform.

**Make your sources open and modifiable.** If only the 'consumer' products of the course (e.g., lecture slides and videos) are published with a license to reuse them, other teachers are forced to take or leave the material as a whole for their course, since any edits would require the huge effort to build sources from scratch and also cut off this teacher from any future improvements of the base material (a hard fork). Therefore, all source files should be made public as well. Furthermore, opening material sources to the public does not only imply public reading access but also the possibility to public contributions to and feedback on the material. A quality control gate has to be implemented in order to ensure that contributions always improve the quality of the material. This can, e.g., be achieved by pull requests in a Git-based system, where suggested modifications are reviewed by members of the core maintainer team.

**Use open licenses.** In order to be able to share material

legally, permissive licenses that allow for modification and redistribution have to be used. The OER community recommends licenses of the Creative Commons organization that were designed for all kinds of creative material (e.g., images, texts, and slides). The approach we are proposing, however, consists of creative material but also of source code that allows third parties to tailor the material. Therefore, also open-source licenses need to be considered: Taking the definition of the Open Source Initiative (OSI) as a guideline, we recommend releasing the material under two different licenses: Source files such as LaTeX, R or Python files should be released under a permissive BSD-like license or a protective GPL or AGPL license, while files such as images, videos, and slides should be released under a Creative Commons license such as CC-BY-SA 4.0.

**Release well-defined versions and maintain change logs.** As development versions of the material will not be overall consistent, it is important to tag versions that can be considered 'stable'. These releases should be easily identifiable and accessible. A change log that lists main changes compared to prior versions should accompany every release.

**Define prerequisites and learning goals of the course.** In order for other lecturers to efficiently evaluate the material for use in their course, it is important to clearly define the scope of the course and its prerequisites, potentially providing references to books or online material which are well-suited to bring students to the desired starting level. Furthermore, each chapter or course unit also needs clearly defined learning goals, so that lecturers can easily select relevant subsets of the material and remix or extend them.

**Foster self-regulated learning.** In our opinion, only active application of newly learned material guarantees proper memorization and deeper understanding. Such application entails example calculations, method applications on toy examples, and active participation in theoretical arguments and proofs. Such exercises are not trivially constructed if one aims at automatic self-assessment to support independent self-study of students. A simple option are multiple-choice quizzes, allowing students to test their understanding after watching videos or reading texts. As students might choose correct answers for the wrong reasons, quizzes should ideally be accompanied by in-depth explanations. Coding exercises, especially important in ML and DS to deepen practical understanding, should be accompanied by well-documented solutions. They can at least be partially assessed by subdividing the required solution into smaller components and defining strict function signatures for each part. Their correctness can now be examined in a piecemeal and step-by-step manner on progressively more complex input-output pairs with failure feedback – pretty much exactly as unit tests are constructed in modern software development.

**Modularization: Structure the material in small chunks.** We recommend structuring the material in small chunks with a very clearly defined learning goal per chunk,

c.f. microlearning and microcontent (Hug, 2006). While microlearning is aimed at enabling more successful studying in smaller, well defined units, we would like to emphasize that such modularization is also highly beneficial from an OSER developer perspective. Highly modularized material can be adapted for different use cases much more easily, and this design principle is analogous to the way good software libraries are constructed. Modularization enables teachers to make changes to specific parts of their course without the need to modify a large set of different chunks (UC4 and UC5). Additional topics and concepts can be plugged in smoothly (UC6) and compiling a smaller, partial course (UC2) is rather convenient and often necessary. Finally, the existing material (or a subset of it) can be used much more easily as a starting point for developing a somewhat different course (UC3).

**Modularization: Disentangle theoretical concepts and implementation details.** For most topics, there is no single best programming language, and preferences and languages themselves evolve quickly over time. To ensure that the choice of programming language does not limit who can study the course, the lecture material should, wherever possible, separate theoretical considerations from coding demos, toolkit discussions and coding exercises. That way, this components can be swapped out or provided in alternative languages, without affecting the remaining material – e.g., an ML lecture with practical variants for Python, R, and Julia, where the latter can be freely chosen depending on the students background. Even more important, this enables a focused, modularized change, if a developer wants to teach the same course via a different programming language.

**Do not use literate programming systems everywhere.** Literate programming systems (Geoffrey M. Poore, 2019; Xie, 2018) provide a convenient way to combine descriptive text (e.g., LaTeX or Markdown) with source code that produces figures or tables (e.g., R or Python) into one single source file. At least for lecture slides, we advise against literate programming systems, and instead advocate using a typesetting system such as LaTeX with externally generated (fully reproducible) code parts for examples, figures and tables to provide modularization of content and content generation. The mixture of typesetting and code language usually results in simultaneous dependency, debugging and runtime problems and can make simple text modifications much more tedious than they should be.

**Enable feedback from everyone.** Feedback for OSER can come from colleagues and other experts, but students and student assistants also provide very valuable feedback in our experience. Providing students the chance to submit pull requests can further improve the material and student engagement. Therefore, we advocate to be open to feedback from all directions and all levels of expertise. Modern VCS platforms like Github provide infrastructure for broad-based feedback via issue trackers, pull requests and project Wikis.

## 4. Using OSER in Blended Learning

High-quality OSER provide an ideal foundation for blended learning scenarios, in which direct interaction with students complements their self-study based on the OSER. Our ideas of how to design an accompanying inverted classroom are based on our experiences from recent years where we have offered several such courses, including an introduction to ML, an advanced ML course, and a full MOOC on a specialization in ML at a platform without paywalls.[1] All the materials are publicly available in GitHub repositories, incl. LaTeX files, code for generating plots, demos and exercises, and automatically graded quizzes for self-assessment.

Even if the goal of the online material is to be as self-contained as possible to optimally support self-regulated learning, an accompanying class – where lecturers and students can be in direct contact – will increase learning success. This class should not consist of repeating the lecture material in a classical lecture, making the videos redundant. Instead, it should use all the online material and add the valuable component of interacting with others – other students and lecturers. The goal of the class should be to encourage students' active engagement with the material by asking and answering open questions, discussing case studies or discussing more advanced topics. It can consist, e.g., of a question and answer session, live quizzes moderated by the lecturer, group work regarding the exercise sheets, and many more. It is key that students are engaged as much as possible here to foster active and critical thinking.

## 5. Challenges and Discussion

**Quality control and assurance of consistency.** A single lecturer should always know the status of their material and can organize changes in any form, without further communication. With a (potentially large) development team, well-intentioned changes can even degrade the quality of the course; consistency of narrative, notation, and simply correctness of edits by less experienced developers have to be ensured. Additionally, it can be a substantial initial effort to integrate existing material of different previous courses from different instructors into a single shared course. Therefore, a quality control process has to be implemented, which generates additional overhead.

**Changed workflow for lecturers.** Developing an online course and teaching in an accompanying inverted classroom changes the workflow of the lecturer. Whereas the material in a classical lecture is presented at fixed time slots during the semester, the online setup allows even more liberal allocation of work time not only for the development, but also for the recording of the material. Furthermore, material can now be iterated in focused sprints and larger parts of well-established lectures can be re-used during the semester

without changes. This can result in large time gains and better control of time allocation for the developer. On the other hand, our experience shows that recording high-quality videos is considerably more time-consuming[2] than a classical lecture in person.

**Technical barrier.** The entire team of developers, from senior lecturers to student assistants, has to work with a much larger toolkit chain that requires more technical expertise. Reducing this entry barrier as much as possible by not over-complicating setups and providing as much guidance from senior developers is absolute key in our experience.

**Enable communication between students and between students and lecturers.** When using the OSER in a full online or MOOC setting, it is important for students to communicate amongst each other and obtain answers from lecturers in order to provide active, positive exchange between all participants and to create a group experience. We think this is a key challenge, and easier to accomplish in a blended learning setup with on-campus sessions. Possible, at least partial remedies are an online forum or a peer-review system for exercises where students give feedback to other students resulting in a scalable feedback system. Especially for the fields of ML and DS, online forums such as Stack Overflow or Cross Validated are widely used and can be reused for lecture questions if threads are properly tagged. This not only reuses existing open tools, but provides the opportunity of exchange with a larger community.

**Granting certificates for online students in MOOCs.** An open issue remains the question if and how (external) students who take a full online course can be granted certificates in some way. Challenges are: (1) Scalability of the grading process for a potentially very large number of students. A possible solution could be assessments by randomly assigned peers in combination with few samples graded by instructors. (2) Preventing fraud and making sure that people answered exam questions on their own. The risk can be mitigated by randomly assigning tasks, asking open questions or assigning more creative tasks for which there is no single correct answer. (3) Designing an evaluation process that evaluates the learning goals of the course.

**Providing solutions.** Solutions should be online and accessible at all times, but focused, unassisted work on solving the exercises has a positive impact on the learning success. It is somewhat unclear whether providing fully worked out solutions encourages students to access these too early.

**Credit assignment.** If a larger group of developers collaborates on a course, it is no longer clear who should get credit for which parts of the material. The quantity and quality of contributions by the different contributors will vary. We recommend a magnanimous and non-hierarchical policy of generous credit assignment that does not emphasize such differences to avoid alienating potential contributors.

---

[1]Details now omitted to preserve double-blind review process.

[2]maybe by a factor of 3-4, personal estimate by one author

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
