# OpenReview forum: "Developing Open Source Educational Resources for Machine Learning and Data Science"
_ecmlpkdd.org/ECMLPKDD/2021/Workshop/TeachML — Submitted to TeachML 2021_

### Official Review · Reviewer_x7Tv · 2021-07-15
**Well written paper about Open Source Educational Resources without significant contribution**

**Rating:** 4
**Confidence:** 4

**Review:**

The authors lay out the importance of Open Source Educational Resources (OSER) and present core principles for their development. These principles are fairly obvious, so the described approach is effectively "Git for teaching resources". For this reason, I can't see any contribution beyond the enumeration of the principles, and consequently vote to reject the paper.

## Strengths:
The paper fits the topic of the workshop, is overall well written and the reader can easily follow the explanations.


## Weaknesses:
Linguistically, the paper contains some incorrect spellings, e.g. "tensorflow" instead of "TensorFlow" or "pytorch" instead of "PyTorch". In some passages the presentation could be improved, e.g. the list of use cases or the in-text enumeration in section 2.
The reference style needs revision: According to the ICML template, the `et~al.' construct should be used for citations with three or more authors. In the references, there are many entries that list over three authors without using this abbreviation. Almost all references are web sources, some of them only pointers to the websites of the mentioned frameworks - these references might be better placed in footnotes.

---

### Official Review · Reviewer_wCZj · 2021-07-15
**Problem well described but no new insights**

**Rating:** 4
**Confidence:** 4

**Review:**


The paper "Developing Open Source Educational Resources for Machine Learning and Data Science" is well structured and easy to read. The authors describe a general approach to make OER materials qualitatively and quantitatively more usable for the ML and DS community. Thereby, the argumentation is followed that, besides the open access to learning materials in the ML/DS area by means of MOOCs, the source files of the learning materials are lacking.
Based on this consideration, ML is defined as particularly suitable for open materials and, based on the definition of OER, the term Open Source Educational Resources is introduced. From this, core principles for OSER are derived and discussed.

The argumentation about MOOCs as OER to OSER is not conclusive. Here, there is a lack of basic literature on OER that defines access to source files as good practice. The introduced term OSER seems unnecessary, whereas the problematic situation of frequently unshared source files is an important problem in the dissemination of OER. The authors note that ML is particularly suited to OER over other disciplines. Here it remains unclear which other disciplines are being referred to here, as the dissemination of OER occurs across disciplines (see https://oerworldmap.org/resource/).

Unfortunately, the Core Principles do not stand out from the general didactic and methodological recommendations for generating learning content. In the discussion section, problems and challenges in the implementation of OER are explained in a comprehensible way. However, these are exactly the problems that have been standing in the way of the dissemination of OER for almost 15 years. Licensing in CC for materials and BSD/MIT for code are generally widespread defaults.

Overall, the article summarizes aspects of didactic, e-learning and the OER community, but unfortunately does not provide any new insights.

---

### Decision · Program_Chairs · 2021-07-21

**Decision:**

Reject

**Comment:**

Thank you for submitting this year to the Teaching ML workshop. The reviewers agree that this paper is not ready for publication. While the statement of the OSER principles are well laid out, there is no clear application of these principles to a specific course or setting. Without a clear application, any insights that the teaching ML community could gain are severely limited.

We encourage the authors to keep up their efforts in the field and act upon the suggestions made. We would love to see a submission from you in the future. We cordially invite you dial in for the workshop itself to be part of our community and make contributions there. We are looking forward to hearing from you in the future.